# Understanding Variational Autoencoders with Intrinsic Dimension and Information Imbalance

**Charles Camboulin**
CY Tech, France[*]

**Diego Doimo**
Area Science Park, Italy

**Aldo Glielmo**[†]
Banca d'Italia, Italy

## Abstract

This work presents an analysis of the hidden representations of Variational Autoencoders (VAEs) using the Intrinsic Dimension (ID) and the Information Imbalance (II). We show that VAEs undergo a transition in behaviour once the bottleneck size is larger than the ID of the data, manifesting in a double hunchback ID profile and a qualitative shift in information processing as captured by the II. Our results also highlight two distinct training phases for architectures with sufficiently large bottleneck sizes, consisting of a rapid fit and a slower generalisation, as assessed by a differentiated behaviour of ID, II, and KL loss. These insights demonstrate that II and ID could be valuable tools for aiding architecture search, for diagnosing underfitting in VAEs, and, more broadly, they contribute to advancing a unified understanding of deep generative models through geometric analysis.

## 1 Introduction

Variational Autoencoders (VAEs) [1] are a central paradigm in the field of representation learning [2] due to the breadth of practical applications [3, 4, 5] and depth of technological developments [6, 7, 8, 9] they have inspired. Their ability to shape useful representations for a wide range of tasks without supervision has made it crucial to interpret their structure and geometry. A first line of work focused on interpreting the latent space and improving the performance of the VAEs by disentangling the latent space features, making each relevant factor of variation of the data dependent on a single latent unit [9]. Since nearest neighbour relations can be used to define meaningful similarities [10, 11], a second line of work [12, 13] described the right notion of distance between datapoints in the latent space and used it to regularise training [14], to improve the clustering [15] and interpolation [16] of the data, and to devise ways to probe and modify their semantics [17].

In this study, we take a different perspective: we describe how two geometrical properties of the hidden representations of the VAE, their Intrinsic Dimension (ID), and their Information Imbalance (II) [18] change through *all* the hidden representations. The analysis of these quantities has proven useful in understanding high-level stages of information processing in convolutional networks [19, 20] and transformers [21]. For instance, in transformers for image generation layers rich in abstract information about data lie between two intrinsic dimension peaks [22] and, in language models, the II can be used to identify blocks that encode semantic information [23]. This work shows that these findings also hold in VAEs despite the very different architecture and training objective. More precisely, we find that **(1)** in VAEs, the ID profiles have two peaks in the middle of the encoder and decoder and a local minimum in correspondence to the bottleneck, and that **(2)** the II identifies a first phase of information compression in the encoder and a second one of information expansion in the decoder, irrespective of phases of expansion and compression of the ID. We also show that **(3)** these geometric features arise only if the bottleneck is larger and the ID of the data and develop the final part of training.

---

[*]This research work was carried out when Charles Camboulin was an intern at Banca d'Italia.

[†]Corresponding author, aldo.glielmo@bancaditalia.it

## 2 Methods.

**Architectures and training details.** We build encoder networks consisting of four convolutional layers with 64, 128, 256, and 256 channels and analyse the geometry of their hidden representations as the size $K$ of the bottleneck increases from 2 to 128. As we train the networks on 32x32 pixel coloured images, the encoders have extrinsic dimensions of 3072 (in the input), 16384, 8192, 4096, 1056, and $K$ (in the bottleneck). The decoder reconstructs the original input by mirroring the encoder architecture and adding a final sigmoid activation to generate the output image. We train each architecture for 200 epochs on the ELBO loss using an Adam optimiser with a weight decay of $10^{-4}$. We train the VAEs on the CIFAR-10 [24] and MNIST [25] datasets. In Sec. 3, we report the ID and II curves on 5,000 images selected from the CIFAR test set and show the results on MNIST in Sec. C of the Appendix.

**Intrinsic Dimension.** The ID of a dataset can be formally defined as the minimum number of variables needed to describe the data with no loss of information [26]. In this work we use the 2NN estimator [27], which computes ID as $N/\sum^N \log \mu_i$, where $\mu_i$ is the ratio of the Euclidean distances between a point $i$ and its second and first nearest neighbours, and $N$ is the number of points in the dataset.

**Information Imbalance.** The II is a recent statistical measure grounded in information theoretic concepts that quantifies the asymmetric predictive power that one feature space carries on another [18]. Specifically, the II *from* space $A$ *to* space $B$ is written as $\Delta(A \to B)$ and is a number between zero and one. If $\Delta(A \to B) = 1$, then the II is at its maximum, meaning that $A$ carries no information on $B$. On the contrary, if $\Delta(A \to B) = 0$, then the II is at its minimum, and $A$ carries full information on $B$. Importantly, the II is *not* a measure of mutual information between spaces. In fact, the II is not symmetric as one space can generally be more predictive of another (for instance, if the data are distributed according to the non-invertible function $y = \sin(x)$, $x$ carries more information about $y$ than vice-versa). $\Delta(A \to B)$ is related to exponential of the conditional entropy $H(c_B \mid c_A)$ of the copula variables $c_A$ and $c_B$ [18, 28] and, it can be robustly estimated by checking how nearest neighbour relationships in space $A$ are preserved in space $B$. In practice, if $r_{ij}^A$ ($r_{ij}^B$) denotes the distance rank of point $j$ with respect to $i$ in space $A$ ($B$) the information imbalance can be computed as

$$\Delta(A \to B) = \frac{2}{N^2} \sum_{i,j|r_{ij}^A=1} r_{ij}^B. \tag{1}$$

To compute IDs and IIs, we use the estimators available in the DADApy package [29].

## 3 Results and discussion

We begin our discussion in Section 3.1 by presenting ID and II profiles of trained VAEs and continue in Section 3.2 by studying how they appear during training. We find that ID and II effectively identify a sharp transition in the VAEs' information processing when the bottleneck size exceeds the input's ID and two distinct phases in the learning process.

### 3.1 Trained Architectures

**The ID identifies a transition through the emergence of a double hunchback profile.** The left panel of Fig. 1 shows how the ID changes from input to output for different bottleneck sizes. We first note that, independently of the architecture, the ID in the encoding part of the network increases from around 20 at the input (layer 0) to approximately 80 before decreasing to roughly match the size of the bottleneck layer (layer 5), creating a distinctive 'hunchback' shape. A similar pattern of expansion and compression is observed in the decoding network, leading to the appearance of a double hunchback curve in the ID. However, the second hunchback curve appears exclusively for VAE architectures with sufficiently large bottleneck sizes, roughly from 32 onwards. Interestingly, the transition occurs when the bottleneck size matches the ID of the input data, which can be read from the figure (at layer 0) to be approximately 30.

**The II identifies a transition through distinct information processing modes.** A similar transition after a specific bottleneck size can be observed through the analysis of II profiles. The central panel

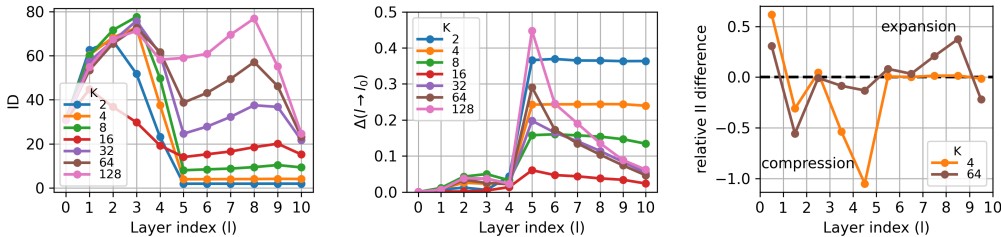

Figure 1: **ID and II of trained architectures.** Left) IDs for different bottleneck sizes $K$. Centre) IIs $\Delta(l \to l_0)$ from layer ($l$) to the input ($l_0$) for different bottleneck sizes $K$. Right) Relative II difference between consecutive layers ($l$ and $l+1$) as a function of layer index $l$. In all panels, the quantities are graphed as a function of the layer index ($l$).

Fig. 1 shows the II from every layer $l$ of the networks to the input layer $l_0$. To start with, we verify that $\Delta(l_0 \to l_0)$ is always zero since the input layer is trivially fully informative on itself. At the other extreme, we see that $\Delta(l_{10} \to l_0)$ starts very high for bottleneck size $K = 2$ and gradually decreases, indicating that the reconstructed images carry limited information about the original images but improve progressively. For reference, the II from the output layer to the input is much higher and around one for any untrained architecture, as shown in Fig.A2 of the Appendix, indicating the absence of any useful information in the output before training. Notably, the II $\Delta(l_{10} \to l_0)$ reaches a minimum for $K = 16$ and then increases for larger sizes (see also the left axis of Fig. A1 of the Appendix), marking a discontinuous change in the nature of the generated images. The transition is also reflected in behaviour of $\Delta(l_5 \to l_0)$, the II from the bottleneck ($l_5$) to the input ($l_0$). Prior to the transition ($K = 2$ to $K = 16$), the bottleneck carries increasingly more information on the input with increasing bottleneck sizes, as indicated by II values decreasing roughly from 0.4 to 0.05. In contrast, after the transition, the bottleneck progressively loses information on the input, as indicated by the II, gradually increasing to over 0.4. These results indicate a qualitative shift in information processing before and after the transition. Before the transition, VAEs predominantly preserve low-level features of increasing detail all the way to the bottleneck, which are, in turn, passed to the output with minimal information processing. After a transition, for sufficiently large bottleneck sizes, the features passed to the bottleneck are no longer directly informative on the input, becoming more abstract and useful for the generation process in the decoder.

The right panel of Fig. 1 provides further support for this hypothesis. It depicts the relative II difference between adjacent layers, defined as $2(\Delta_{l,l+1} - \Delta_{l+1,l})/(\Delta_{l,l+1} + \Delta_{l+1,l})$ where $\Delta_{l,l'} = \Delta(l \to l')$. The quantity is plotted as a function of the layer index and measures the relative information content of consecutive layers. A value above zero indicates that the $l + 1$ layer is more informative than the $l$ layer, signifying an expansion of information content. Conversely, a negative value indicates a compression. The graph clearly illustrates that while both $K = 4$ and $K = 64$ undergo a compression in the encoder, only the $K = 64$ architecture also undergoes an expansion in the decoder as the small bottleneck size of the $K = 4$ architecture inhibits such expansion. Lastly, a transition is also visible by comparing the relative information content of neighbouring architectures (e.g., $K = 2$ vs $K = 4$ or $K = 64$ vs $K = 128$) for fixed layers, as done in Fig. A1 of the Appendix.

## 3.2 Training dynamics

**The KL loss identifies two phases of training.** The sharp transition described in the previous section is also reflected in qualitatively different training dynamics before and after a specific bottleneck size. The left and centre panels of Fig 2 display the the FID loss [30] and the KL loss, respectively. While the FID loss decreases monotonically as training progresses for all architectures, the KL loss exhibits a differentiated behaviour. Specifically, only for architectures with sufficiently large bottleneck dimension the KL loss goes through two distinct phases. In the first phase, roughly until epoch 10, the KL loss *increases*, while it decreases in the second phase until the end of training.

**The II identifies two phases of training.** A similar dynamic can be observed in the right panel of Fig. 2, which shows $\Delta(l_{10} \to l_0)$, i.e., the II from the output layer to the input, as a function of the training epoch. The figure illustrates that, after a minimally large bottleneck size, the II

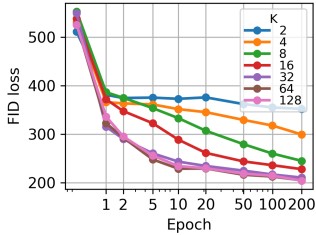 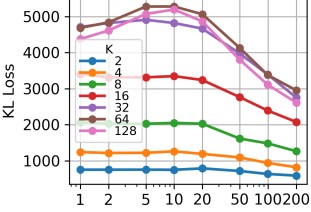 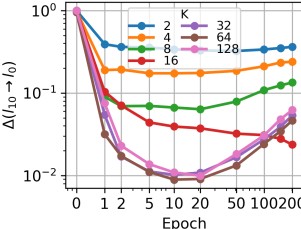

Figure 2: **Losses and II during training.** Left) FID loss on test set. Centre) KL Loss on test set. Right) II from last layer ($l_{10}$) to the input ($l_0$). In all panels, the quantities are graphed as a function of training epoch and for architectures of increasing bottleneck sizes (denoted by different colours).

dynamic exhibits two phases that closely parallel the two phases of the KL loss during training, with the II decreasing sharply up to epoch 10 and then increasing gradually for the rest of the training.

We can interpret the first phase as a rapid 'fitting' phase to the data and the second phase as a more gradual 'generalisation' phase where the bottleneck becomes more Gaussian (leading to a decrease in KL loss), and the output discards the details of the input data that are not useful for the generative task (leading to an increase of $\Delta(l_{10} \to l_0)$).

**The double hunchback ID profile emerges in the second phase of training.** Finally, Fig. 3 shows the evolution of the ID across layers during training for the $K = 64$ architecture. The graph illustrates that the evolution of the ID curve also mirrors the two phases of training, as the characteristic double hunchback shape emerges only after epoch ten and during the second training phase.

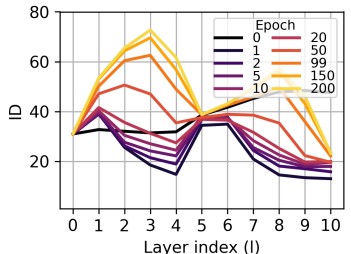

Figure 3: **ID during training.** ID as a function of layer index for different epoch numbers for $K = 64$.

## 4 Conclusions

In this study, we explored the use of geometric tools –the ID and the II– to analyse the internal representations and training dynamics of Variational Autoencoders. We identify a sharp transition in the behaviour of VAEs as the size of the bottleneck layer increases above the ID of the data, leading to the emergence a double hunchback curve in the ID profile and to a qualitatively different information processing mechanism as measured by the II. Furthermore, we find that architectures with sufficiently large bottleneck undergo two distinct phases in the training dynamic.

**We envisage two practical implications of our findings.** First, they can inform architecture search since a well-performing VAE network can be expected (and measured) to exhibit a double hunchback ID curve and an II information processing involving a compression and a later expansion. Second, monitoring ID and II during training can provide valuable insights into the learning process, potentially enabling diagnostic tools for avoiding underfitting and improving training.

More generally, the geometric analysis we propose can be useful for robustly and nonparametrically compare architectures, when dealing with high-dimensional spaces. For example, the double hunchback shape we observe in VAEs closely mirrors the findings of [22] in their analysis of Transformers trained on ImageNet, suggesting this shape may be a common feature of a large class of deep generative networks. Furthermore, the two training phases we identify resemble the two phases proposed in the Information Bottleneck (IB) theory of deep learning [31, 32]. However, while previous studies on the IB theory and its application to neural networks [33] have faced challenges due to the difficulty of accurately estimating mutual information [34, 35], our approach circumvents these issues by employing distance-based geometric tools that allow reliable studies of the information processing dynamics even in high-dimensional hidden representations.

Further work will involve validating these findings in other architectures and establishing a more rigorous connection between the geometric measures and information-theoretic concepts.

## Code availability

The code we used to train the models and to compute Intrinsic Dimensions and Information Imbalances is available at https://github.com/bancaditalia/Understanding-Variational-Autoencoders-with-Intrinsic-Dimension-and-Information-Imbalance.

## Disclaimer

The views and opinions expressed in this paper are those of the authors and do not necessarily reflect the official policy or position of Banca d'Italia.

## Acknowledgments and Disclosure of Funding

We thank Oliver Giudice (Banca d'Italia) for helpful discussions.

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

# Appendix

## A   Network architecture and training details

We build encoder networks consisting of four convolutional layers, each with 4×4 kernels, a stride of 2, a padding of 1, and number of channels of 64, 128, 256 and 256 respectively. Each convolutional layer is followed by batch normalisation and Leaky ReLU activation. After the final convolutional layer, the output is flattened into a vector, which is passed through fully a connected layer to compute the mean and log-variance of the latent space of various dimension $K$ depending on the architecture from 2 to 128. As we train the networks on 32-pixel colour images, the encoders have extrinsic dimensions of 3072 (in the input), 16384, 8192, 4096, 1056 and $K$ (in the bottleneck). The decoder reconstructs the original input by mirroring the encoder architecture, and adding a final sigmoid activation to generate the output image. We build our models using PyTorch [36] and train each architecture for 200 epochs on the ELBO loss using an Adam optimiser with a weight decay of $10^{-4}$. We used the CIFAR-10 dataset [24] for the experiments presented here, consisting of 60,000, across 10 classes representing various objects (e.g., airplanes, cars, and animals). The dataset is divided into a training set of 50,000 and a test set 10,000. For training, we used 0.8 of the training set, with the remaining 0.2 reserved for validation. We performed training on a single T4 GPU with 16MB of RAM using batch size of 256 and one run with 200 training epochs took around 40 minutes. For the computation of ID and II curves we selected 5,000 images from the test set. In the Appendix, we report the results obtained by training the same architectures on the alternative MNIST dataset [25].

# B CIFAR10 extra results

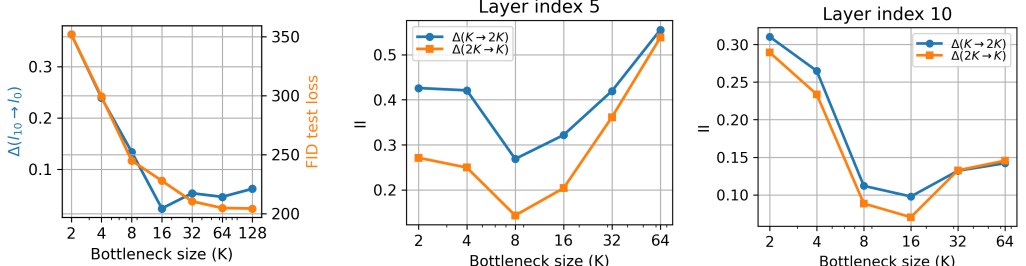

Figure A1: **II and FID, and IIs between different architectures.** Left) II from the output to the input (left axis) and FID test loss (right axis) for increasing bottleneck sizes ('latent dimensions'). The panel clearly shows a transition in the nature of the output layer of VAEs as the bottleneck surpasses a critical value. This transition is not paralleled by an increase in the test error as measured by the FID loss. Centre and Right) The II curves for layer indices set to $l_5$ (bottleneck) and $l_{10}$ (output) and measuring the imbalances across different architecturs $\Delta(K \to 2K)$ and $\Delta(2K \to K)$ where $K$ is the size of the bottleneck. The two panels show that a transition is present in the nature of the bottleneck layer (layer 5) and output layer (layer 10) when the bottleneck size of VAEs surpasses a value of $K = 16$.

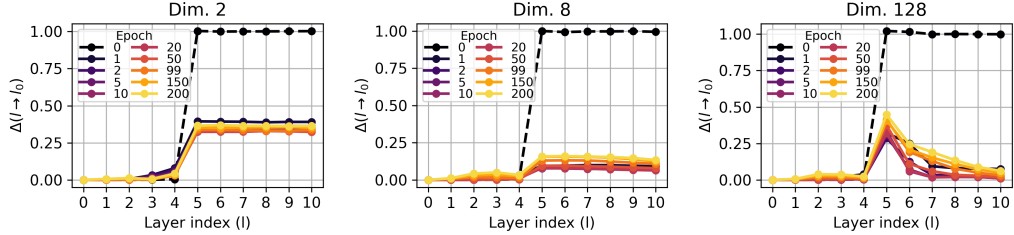

Figure A2: **II curves during training.** II from every layer ($l$) to the input ($l_0$) as a function of the layer index for increasing epochs during training and for architectures with different bottleneck sizes (2, 8 and 128, here referred to as 'Dim.'). The figures show that before training (dashed black curve) the II is zero up to the bottleneck and one afterwards, indicating full information in the encoder and no information in the decoder. The figures also illustrate the difference in II before and after the transition at $K = 16$ .

# C    MNIST Results

We observe similar tendencies on MNIST, although with some key differences due to the simpler nature of the dataset. The characteristic (double) hunchback shape remains, but the overall ID and II range is smaller comparatively, reflecting the reduced feature complexity in MNIST compared to CIFAR-10. In particular, the ID in the encoder starts from around 10 at the input layer and peaks at around 25, before compressing to match the size of the bottleneck layer. And during expansion in the decoder the ID goes up to 60 for the highest bottleneck size, largely exceeding the ID peak observed in the encoder. Another important difference lies in the fact that the critical bottleneck size for the transition is $K = 8$ and not $K = 16$. As the ID of the input image is smaller for MNIST and around 10-12, this finding supports the hypothesis that the transition emerges after the bottleneck has matched the ID of the input data.

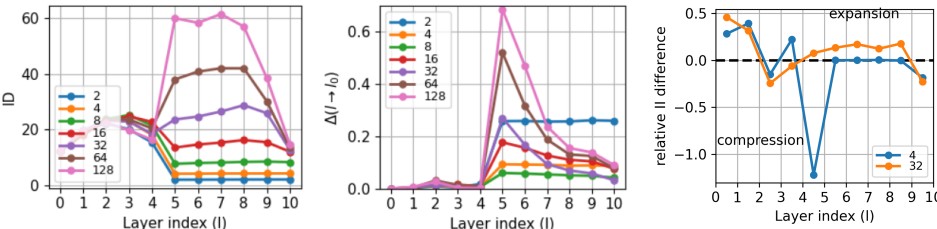

Figure A3: **ID and II of trained architectures.** Left) IDs for different bottleneck sizes (different colours). Centre) IIs $\Delta(l \to l_0)$ from layer $(l)$ to the input $(l_0)$ for different bottleneck sizes (different colours). Right) Relative II difference between consecutive layers ($l$ and $l + 1$) as a function of layer index $l$ for bottleneck sizes of 4 and 32. In all panels, the quantities are graphed as a function of the layer index $(l)$.

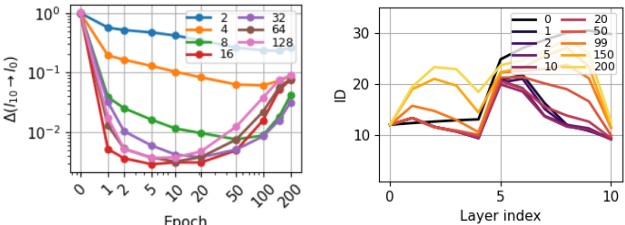

Figure A4: **II and ID during training.** Left) II from the last layer ($l10$) to the input ($l_0$) as a function of the epoch number and for different bottleneck sizes (different colours). Right) ID as a function of layer index for different epoch numbers (different colours) and for latent dimensions 4, 32 and 64 respectively.

