# OpenReview forum: "Understanding Variational Autoencoders with Intrinsic Dimension and Information Imbalance"
_NeurIPS.cc/2024/Workshop/UniReps — UniReps_

### Official Review · Reviewer_XFrV · 2024-09-28
**ID and II in VAEs**

**Rating:** 6
**Confidence:** 3

**Review:**

The paper analyzes the behaviors of intrinsic dimension, information imbalance and KL divergence.

Strength: 1. The presented interplays of the measures are interesting. 2. The paper may open revenue for new research.

Weakness: 1. In its current state, the results are largely empirical. It would be interesting if theoretical results are provided. 2. It is not clear how reliable is the employed intrinsic dimension estimator; see [1] for some results, which shows that 2NN may not be reliable in high dimension.

[1] Intrinsic dimensionality estimation using Normalizing Flows, Horvat and Pfister, https://proceedings.neurips.cc/paper_files/paper/2022/hash/4f918fa3a7c38b2d9b8b484bcc433334-Abstract-Conference.html

---

### Official Review · Reviewer_ghMm · 2024-10-07
**The empirical analysis of hidden VAE representations conducted through geometric properties, specifically information imbalance and intrinsic dimension.**

**Rating:** 7
**Confidence:** 2

**Review:**

Strengths :

* The idea is based on two previous works with different architectures, and the paper presents a preliminary results for the analysis of the hidden VAE representations through geometric properties. Previous work is cited adequately.
* The metrics used are thoroughly explained, and the claims are well-supported by the experiments. Additionally, the results are evaluated. However, the approach should be tested on different datasets to further validate its effectiveness.
* The paper is overall well written, with clear implementation details for reproducibility.
*  While the approach is not entirely novel, its application to VAEs has the potential to improve the understanding of the architecture, as stated in the paper, providing a new perspective.

Weaknesses / Suggestions:
* The choice of geometric properties (II and ID) could benefit from further justification, rather than referencing prior works that used them in different architectures and tasks.
* The paper does not sufficiently discuss the limitations of the work.
* The experiments on the MNIST dataset are not sufficient to fully support the claims in the paper. Conducting experiments on a different dataset could improve the validation of the methods.

---

### Decision · Program_Chairs · 2024-10-10

**Decision:**

Accept

**Comment:**

In light of the positive reviewers' feedback and relevancy of the submission, we are pleased to accept this paper for presentation at UniReps 2024. We kindly ask the authors to incorporate the reviewers' suggestions and feedback in the final camera-ready version of the manuscript.